# Investment case approach for equitable access to maternal neonatal and child health services: Stakeholders' perspective in Nepal

Janak Kumar Thapa [1]*, Doris Stöckl[2], Raj Kumar Sangroula[3], Asha Pun[4], Meena Thapa[4], Mahesh Kumar Maskey[5], Maria Delius[6]

**1** CIH[LMU], Center for International Health, Ludwig-Maximilians-University, Munich, Germany, **2** Helmholtz Zentrum Muenchen, University Hospital, LMU Munich, Munich, Germany, **3** Nepal Public Health Research and Development Center (PHRD Nepal), Kathmandu, Nepal, **4** UNICEF, Kathmandu, Nepal, **5** Nepal Public Health Foundation, Kathmandu, Nepal, **6** Department of Obstetrics and Gynecology, University Hospital, LMU Munich, Munich, Germany

* janakthapa7@gmail.com, Janak.Thapa@lrz.uni-muenchen.de

**Data Availability Statement:** All relevant data are within the paper and its Supporting information files.

## Abstract

### Background

Investment Case is a participatory approach that has been used over the years for better strategic actions and planning in the health sector. Based on this approach, a District Investment Case (DIC) program was launched to improve maternal, neonatal and child health services in partnership with government, non-government sectors and UNICEF Nepal. In the meantime, this study aimed to explore perceptions and experiences of local stakeholders regarding health planning and budgeting and explore the role of the DIC program in ensuring equity in access to maternal and child health services.

### Methods

This study adopted an exploratory phenomenography design with a purposive sampling technique for data collection. Three DIC implemented districts and three comparison districts were selected and total 30 key informant interviews with district level stakeholders and six focus groups with community stakeholders were carried out. A deductive approach was used to explore the perception of local stakeholders of health planning and budgeting of the health care expenses on the local level.

### Results

Investment Case approach helped stakeholders in planning systematically based on evidence through collaborative and participatory approach while in comparison areas previous year plan was mainly primarily considered as reference. Resource constraints and geographical difficulty were key barriers in executing the desired plan in both intervention and comparison districts. Positive changes were observed in coverage of maternal and child health services in both groups. A few participants reported no difference due to the DIC program. The participants specified the improvement in access to information, access and

**Funding:** The authors received no specific funding for this work.

**Competing interests:** The authors have declared that no competing interests exist.

**Abbreviations:** FCHV, Female Community Health Volunteer; MBB, Marginal Budgeting for Bottlenecks; MNCH, Maternal neonatal and child health; NHSSP, Nepal Health Sector Support Program; PNC, Post-natal care; SBA, Skilled Birth Attendant; SDG, Sustainable Development Goal; UNICEF, United Nations Children's Fund; USAID, United States Agency for International Development; VDC, Village Development Committee; WHO, World Health Organization.

utilization of health services by women. This has influenced the positive health care seeking behavior.

## Conclusions

The decentralized planning and management approach at the district level helps to ensure equity in access to maternal, newborn and child health care. However, quality evidence, inclusiveness, functional feedback and support system and local resource utilization should be the key consideration.

## Introduction

Nepal reduced maternal mortality from 800 in 1990 to 213 in 2015 per 100000 live births and is progressing towards achieving the Sustainable Development Goal (SDG) target of 70 per 100000 live births by 2030 [1]. These improvements are results of focused strategies like addressing direct causes of maternal deaths, increasing skilled birth attended deliveries, promoting of institutional delivery and strive towards ensuring universal access to basic maternal and child health services [2, 3]. Nepal has launched various targeted programs, policies and strategies with the help of international organizations since the 1990s. Such efforts are aimed at reducing barriers on the demand side as well as on the supply side to ensure proper service provision, delivery and utilization. The National Safe Motherhood Program in 1997 [4], the National Safe Motherhood Plan 2002–2017 [5], the SBA policy in 2006 [6], the National Neonatal Health Strategy 2004 [7], and the Community Based Newborn Care Program in 2007 are some of the major programs, plans and policies to be highlighted. Notwithstanding these improvements, equity gaps in maternal and neonatal service utilization across geography, ethnicity and economic status have always been existing and is clearly evident from the findings of the Nepal Demographic Health Surveys of 2006, 2011 and 2016 [8–11]. However, the past efforts still seem to be insufficient considering access to maternal and child health services particularly for rural and poor people. Nepal was found to have 0.67 doctors and nurses per 1,000 people, which is less than the WHO recommendation of 2.3 doctors, nurses and midwives per 1,000 people [12]. Furthermore, in the primary health care system, which is the main health service provider, the existing critical knowledge and capacity gaps prevent contribution to health plans and budgets [13]. Health system constraints often impact most negatively on disadvantaged populations, resulting in poor access to quality health care among vulnerable groups [14].

In order to add on to the existing district health system, particularly in the planning and budgeting process related to maternal, child and neonatal health programme, the District Investment Case program (DIC) was launched in 21 district of Nepal from 2011 to 2016. Low developed (based on Human Development Index (HDI) 2011 value) districts from the UNICEF Nepal working areas were selected comprising different geographical terrains. The Investment Case (IC) approach uses concepts from the 'Tanahashi model' and the 'bottlenecks analysis framework' to identify the current obstacles to health care access and determines the costs and impact of potential interventions to improve performance and overall equity [13]. The IC analysis is based on the distinctive exploration of constraints of health care services from the supply side (availability of essential commodities, availability of skilled human resources, physical accessibility) and from the demand side (initial utilization of services, continuity of services and effective coverage) (modified from Tanahashi) [15]. The main actors for

implementation of the DIC programs were District Health Officers, Public Health Nurses, Local Development Officers, Village Development Committee (VDC) secretary, frontline Health workers from the community level, Female Community Health Volunteers (FCHVs) and so on. Implementation of DIC is usually a five-step process which starts from advocacy with the government, selection of interventions (tracers), data mapping and collection, data validation, bottlenecks analysis and strategies and ends with presenting results (cost and impact) to the stakeholders and creating a buy-in for implementation. After the strategies are developed with consultation with the local stakeholders, simulating outcomes including costs using Marginal Budgeting for Bottlenecks (MBB) tool. The whole process is grounded in evidence starting from identifying constraints to calculating impacts [16].

In the DIC program the aim was to identify inequities in health service coverage and to design targeted interventions. Existing data suggest that the IC approach helps improving the situation of maternal and child health in Nepal and other countries [13, 14, 17]. This qualitative study aims to understand the perception of local stakeholders of health planning and budgeting of the health care expenses on local level. Beyond this, we explored the local stakeholders' evaluation of the effectiveness of the IC approach in having influence on the planning and budgeting process to ensure equitable access to Maternal neonatal and child health (MNCH) services in Nepal.

## Material and methods

### Study setting

The study was conducted in from June to December 2018 in purposively selected districts based on HDI values, three intervention and three comparison districts. The selection of the districts ensured one district each from mountain, hill and plain areas in both the intervention (Bajhang, mountain; Baitadi hill; Parsa plain) and the control (Darchula mountain; Sindhuli hill; Bara plain) group. The intervention areas were DIC program implemented districts while control areas were DIC non-implemented districts with similar geographical characteristics. Before the implementation of the DIC program all study districts shared similar health service coverage indicators [18].

### Study design

This research investigated stakeholders' perceptions and experiences on the Investment Case approach to ensure equitable access to MNCH services in Nepal. The phenomenographic research approach was used for exploring their perceptions as well as experiences [19]. The following subsections explain the procedure in detail.

**Phenomenography** is a qualitative research method, that takes into account, that researchers hold subjective assumptions regarding the interpretation of knowledge. Within this methodology the conception of phenomena is relational, and the world is constructed differently according to the subjective viewpoint and the context [20]. This research approach here is not to discern stakeholders' experiences of the phenomenon as such; rather it stresses identifying variations in the ways of experiencing the phenomenon presented in qualitatively different categories. The aim of using phenomenography in this study was to find a set of themes derived from the stakeholders' understanding regarding the investment case approach to ensure equitable access to MNCH services.

Each topic reflects individual experience by the investigators and describes inter-individual similarities and dissimilarities [21]. Combining these themes together demonstrates a structure, called *outcome space* that describes the different ways and relationships through which a researcher interprets how a phenomenon is experienced at a collective level [22]. Within the

**Table 1. Details of the participants.**

| Participants | Participants' details |
|---|---|
| District (public) Health officers | District level health managers who oversee all the health activities in the entire district with main role in implementation of DIC in the district |
| Public Health Nurses | Their roles were to monitor the tracers of DIC and supervision of the community level stakeholders |
| Frontline health workers | Their roles in DIC were to directly involve in implementation DIC in community level and monitor the indicators |
| Local Development Officers | Chiefs of the Local Development Office, in the district who coordinate and oversee the entire development activities of the district including health care related activities and budgeting |
| The secretaries of the village development committees | Their roles were to allocate the budget for health related activities in community level and monitor the progress of DIC |
| Female Community Health Volunteers (FCHVs) | The role of FCHVs is to advocate healthy behavior by mothers and community people. |
| Mothers of under 2 years' children | They are the main beneficiaries group for DIC |

premise of this research, we explored the perception and experiences of stakeholders. Therefore, we followed a phenomenographic research strategy to determine sample size, collect data, and analyze data to grasp the *outcome space*.

**Study participants.** Both the intervention and comparison districts had equal number of Key Informant Interviews (KIIs) and Focus Group Discussions (FGDs). The study participants were key stakeholders from different levels of the health care system as shown in Table 1.

A purposive sampling technique was used to select the participants and to ensure the required level of variation among the stakeholders' experiences and, consequently, among their ways of perceiving the District Investment Case program. We followed three main principles to maximize the variations among the participants in program districts: (a) each selected stakeholder had some experience or was part of a District Investment Case program at some point for at least six months during the implementation phase to ensure a maximum in-depth understanding of the issue in the intervention districts. However, the degree of experience among them differed. In the comparison districts, the following two criteria were considered; (b) stakeholders recruited were from different levels or tiers in the health system; (c) participants selected were from different geographical regions and administrative regions of the country to maintain enough diversity in phenomenographic study.

## Data collection

For the collection of data, 3 FGD facilitators with Bachelor's in Public Health degree with 2 years' experience and three note takers with Bachelor's degree in Public Health with 2 years' experience in qualitative research were recruited. The same teams were involved in taking Key-informant interviews too. Three days' intensive qualitative research-related training was given to the team by the lead researchers and other co-researchers who were experts in qualitative research. To explore the perception of local stakeholders of health planning and budgeting of the health care expenses on the local level, we conducted 15 KIIs and 3 FGDs each in intervention districts comparison districts. The duration of FGDs ranged from 50 and 60 minutes, while the KIIs ranged from 45 to 50 minutes. The key-informant interviews took place in the District Health Offices, Local Development Offices, Health Posts, Districts Hospitals of the respective districts. The FGDs took place in the room of health institutions or community buildings. FGD and IDI guidelines were prepared by the lead researcher with the assistance of co-researchers that included investment case approach like facilitators and barriers in

planning, implementation and evaluation of the process. Pre-testing of the guidelines was done in one IC implemented district and one non-implemented district which were not part of the study. Some modifications were made after the pre-testing. Data collection was carried out by the facilitators in the local language. All the participants responded and agreed to participate in the study. Notes of all the FGDs and KIIs were taken and also were audio recorded. Confusing and important questions were probed and repeated in order to explore their experience and perception of the IC approach. All the data from FGDs and KIIs were transcribed from the audio recordings and notes. The collected data were checked by the lead researcher and other co-researchers for their accuracy and quality for ensuring descriptive and interpretation validity. In case of missing data, the participants were followed up on the next day. The transcripts were translated into English by the first author and third author.

### Data analysis

Data were analyzed manually and thematic analysis was used [23]. The translated transcripts were read multiple times by the first author and third author multiple times for being familiar with the data. A deductive approach was used to explore the perception of local stakeholders of health planning and budgeting of the health care expenses on the local level. Firstly, all the FGDs and KIIs were compiled and were labeled according to the types of the participants and districts. After reviewing each FGDs and KIIs, the data were organized and were written in Microsoft word in Nepali language. All the transcriptions were translated into English language and were reviewed to understand the meaning of the content by the first and third author. Coding of the data were done by first and third author. Both the authors coded the one FGD each from intervention and control district independently. Similarly, 5 KIIs from the intervention and 5 KIIs from comparison districts were also coded by the first and third authors. Then the codes identified from the texts were then discussed among all the authors, and the final codes were decided. All the remaining transcripts from FGDs and KIIs were then coded using the final codes. The codes with similar meanings were then put together to form a common theme. The themes were then reviewed by all the authors and were finalized. Important verbatim quotes were also included in each theme. All data were organized according to the themes and were summarized according to the pattern of findings for all FGDs and KIIs. The final stage was done in an iterative pattern in regular discussion among all the authors. The list of codes and themes of the study are given in Table 2.

**Table 2. Themes and categories for data analysis.**

| Themes | Codes |
|---|---|
| 1. Perception of health service planning in the district<br>2. Perception of the resource allocation in health services<br>3. Perception of the plan implementation<br>4. Perception of equity in the access to MNCH services | Steps and process of planning involved<br>Evidence based planning<br>Role of IC approach in planning and budgeting<br>Participatory planning<br>Community Participation<br>Influencers and hindrances in planning<br>Ownership<br>Improvement in Coordination<br>Budget allocation process<br>Partnership with external partners<br>MNCH Program implementation<br>Monitoring and supervision<br>Improvements in MNCH indicators<br>Human resource availability<br>Equity<br>Quality of health services |

### Ethical considerations

The ethical approval was obtained from the Ethical Review Board (ERB) of Nepal Health Research Council (reference no. 1296) and the Centre for International Health LMU (CIH$^{LMU}$). Written informed consent was obtained from the participants before the interview. The principal researcher has securely stored the signed informed consent forms.

## Results

The baseline characteristics of the participants like age, sex, religion, ethnicity, marital status and education level are shown in Table 3.

### a) Perception of health service planning in the district

In all districts, the health service planning was executed by a bottom-up approach, starting from the grassroots' level, to the ward level to the Village Development Committee (VDC)

**Table 3. The baseline characteristics of the participants.**

| Characteristics | KII | | FGD | |
| --- | --- | --- | --- | --- |
| | Frequency | Percentage | Frequency | Percentage |
| **Age** | | | | |
| Less than 30 | 3 | 10 | 31 | 59.7 |
| 30–40 | 10 | 33.3 | 16 | 30.7 |
| 40–50 | 11 | 36.7 | 4 | 7.7 |
| More than 50 | 6 | 20 | 1 | 1.9 |
| **Sex** | | | | |
| Male | 21 | 70 | 0 | - |
| Female | 9 | 30 | 52 | 100 |
| **Religion** | | | | |
| Hindu | 22 | 73.3 | 34 | 65.4 |
| Buddhist | 7 | 23.3 | 14 | 26.9 |
| Christian | 1 | 3.3 | 4 | 7.7 |
| **Ethnicity** | | | | |
| Brahmin/Chhetri | 13 | 43.3 | 19 | 36.6 |
| Janajati | 9 | 30 | 16 | 30.8 |
| Dalits | 6 | 20 | 15 | 28.8 |
| Madhesi | 2 | 6.7 | 2 | 3.8 |
| **Marital status** | | | | |
| Married | 29 | 96.7 | 50 | 96.2 |
| Unmarried | 1 | 3.3 | 0 | 0 |
| Widow/widower | 0 | | 2 | 3.8 |
| **Education** | | | | |
| Read and write only | 0 | 0 | 4 | 7.7 |
| Primary | 0 | 0 | 14 | 26.9 |
| Lower secondary | 0 | 0 | 22 | 42.3 |
| Secondary | 0 | 0 | 10 | 19.2 |
| Higher secondary | 12 | 40 | 2 | 3.8 |
| Bachelor level | 18 | 60 | 0 | 0 |

The responses from the FGDs and KIIs discovered four themes. The findings are presented according to the themes generated, reflecting the stakeholders' perception and their experience regarding the health planning process and the DIC program.

level to be approved at the district level. The planning process considered the plan of the previous year and the available local resources. Though the planning in both study groups (intervention and non-intervention) was based on the locally perceived problems and the resources available on-site, in the intervention districts strategies were developed through planning workshop at the district level. No difference was observed in the grass- root level planning. Problems made out at the different organizational levels were taken to the next higher level to be discussed there. Intensive community participation during the planning was wanted, with the involvement of the ward citizen forum. Improved participation of women disabled and of people of the lowest caste (Dalit) was tried to be reached. However, there was a common voice of women in both study groups, that they or their family members are often not consulted during the village level annual MNCH planning.

> "*No, I don't know about this annual planning. We are not consulted during the process. Leaders do that themselves.*"
>
> –a 48 years' old FCHV from intervention district

In some cases, Female Community Health Volunteer (FCHVs) were involved in the planning process, but not included during the budget allocation. One FCHV from the comparison area explains:

> "*They used to call us in the planning process but while allocating budget they do it by themselves*".
>
> –A 41 years' old FCHV from comparison district

The changes that could be observed at the district level in the intervention areas were quite impressive, as felt by many stakeholders. The attitude towards the annual health planning was much more positive. Many study participants felt that the involvement of political leaders in the DIC workshop had positive influence on the annual planning. This was not observed in the comparison areas. Stakeholders often used the past year's plan as a reference, which was approved then with only certain modifications. In the comparison districts the stakeholders felt lack of ownership of their plans and perceived that the planning was driven from the national level.

> "*We don't bother much about annual planning. Budget from central level is with certain modification and we accordingly adjust our past year plan. . .. Of course, we look at health indicators and what target was given by the central level and adjust our plan accordingly. But most of the time our activities are guided by central level. Even if we send our plan and budget, most of the activities and budget heading are removed from there.*"
>
> - A 35 years' old Health Post In-charge from comparison district

In the study, poor linkage of the planning on VDC level with the DIC program was observed. Within the DIC approach, health indicators were obtained on the community level through various records and should have been used for the district level planning. Unfortunately, the annual plan developed at Village Development Committee (VDC) level, which is directly submitted to the District Development Committee (DDC) was not considered too much extent. The stakeholders in some districts, both intervention and comparison, faced problems in coordinating with the DDC, particularly to incorporate their plans in the district red book.

Technical staff and frontline health workers in the intervention districts expressed that the IC planning process was systematic and continuous, it was evidence based and not perception based, and the plans developed were focused to achieve the targets set in the IC districts. In contrast, asking stakeholders in the comparison districts, whether they felt the need of a DIC type program during annual planning, most of them agreed. Many health workers from the comparison districts lacked knowledge about the health planning process.

One of the barriers opposing the planning process reported in the comparison districts was the lack of clarity among the stakeholders about their responsibilities. On the other side, in the intervention districts more clarity regarding responsibilities was observed. From the district level stakeholders, concerns about the technical challenges for planning and implementation of the DIC were raised. "*It is a good approach for using data, analyzing data, and using the data for proper planning. However, it needs a lot of resources to operate it. It is somewhat complicated and hard to understand different level of stakeholders.*"—A 44 years' old District Health Officer from intervention district.

## b) Perception of the resource allocation in health services

Besides governmental resources, UNICEF, USAID were major resource providers in the intervention areas while SUHAARA and Plan International in the comparison districts. In both the intervention and comparison districts, the health workers and district focal persons were coordinating with those partners. They were also involved in the planning and the implementation of the programs/ projects. For the allocation and segregation of the budget, a bottom-up approach with a vigorous discussion of all stakeholders at all levels was carried out. The plans requiring big budget and human resource related support were transferred to the national level and the low to medium budget activities were approved at the district level. This was common in both groups.

In the intervention districts after the IC implementation, prioritization of the allocation of budget for health services under different headings was found, which was further taken to the council for approval. Also, MBB tools were used at the district level in case of intervention while no such standard or similar tools were used in the comparison districts.

According to a 46 years' old District Health Office (DHO) from an intervention district, "*. . .. Before DIC, the budget separated for health by all 68 VDC was just NRs. 5 million (US $ 4500). Currently, the amount separated for health programs by these VDCs has increased to NRs. 1 crore 35 million (US $ 120,000).*" On other side, there was no significant difference in allocating budget for health among comparison districts.

## c) Perception of the plan implementation

The district and community-level health stakeholders reported that the implementation of health programs runs per annual plan. A similar level of experiences and perceptions was observed in both groups regarding the plan implementation. Our focus was to observe how to overcome the implementation challenges. In the intervention districts, there was a support system that continuously guided and supervised their implementation and provided technical help. The district health officers as well as health workers mentioned that the IC approach has played an effective role in not only systematic planning but execution of health activities as well.

"*Enhanced collaboration and coordination among various stakeholders, such as political parties, health workers, VDCs and others to address these health issues has made the*

*implementation effective and easier. Otherwise, it is very hard to convince political leaders for investment in in health.*"

- A 34 years' old public health nurse from intervention district

Similarly, the Local Development Officer (LDO) and the VDC secretary both expressed that, the IC approach had successfully made all stakeholders accountable and supported to meet the health indicators considerably. The implementation process has been effective, and the local stakeholders are positive towards it. They further added that UNICEF has supported strengthening health plans for the MNCH sector at the district level to reach optimal health status by addressing deep-rooted problems and by supporting the community level planning.

In contrast, in the comparison districts there was a lack of an external support system. So, the implementation challenges were handled within the capacity of the existing health system structures. Of course, there was conventional support from the central and the regional health team. Proper monitoring and supervision of the program were not done even in the intervention districts though they had received support from the DIC team. Also, frequent transfer of staff was the main reason reported to be an obstacle in the proper implementation of the program.

The need for the strategic location of the I/NGOs in geographically remote areas and timely fulfillment of their commitment needs to be assured with strong advocacy from the national and the community level governments. This was unanimously mentioned by all stakeholders. For effective implementation, the stakeholders dominantly expressed that there is a need for ownership among community people and stakeholders for its successful implementation.

Most of the health workers suggested that there was the need to stress the government's responsibility to implement local level plans in the national MNCH plans and to provide budgetary assistance for VDC wise programs. On the government level public health activities should be increased like awareness programs and customization of Information, Education and Communication (IEC) materials in local languages. For effective program implementation and evaluation of the strategy, DHOs and Public Health Nurses (PHNs) suggested to focus stronger on the implementation, the monitoring and the supervision of the action plans set out during the IC-approach workshops.

In case of comparison districts the lack of monitoring, supervision and a functional feedback system was reported by the stakeholders.

"*there are a support system and feedback but in practice it doesn't work. Constructive feedback is not working well.*"

- A 46 years' old VDC secretary from comparison district

## d) Perception of equity in the access to MNCH services

Stakeholders from all districts reported to have ameliorated the maternal and child health situation, including easier access compared to past. Such improvements were mainly led or driven by national interest rather than locally grown. However, in the case of the intervention districts the participants reported to have locally developed innovative ideas such as using flag systems (unique color flags for households with golden 1000 days' mother) to track pregnant and lactating mothers to ensure no one is left behind. In most places of the intervention districts, renovation and revitalization of primary healthcare outreach clinics, reactivation of mothers groups, investment in health infrastructure were more frequently observed compared to the non-intervention districts.

"*When a child is born in the community, we provide the child's naisargik adhikar (right by birth) like vaccinating the baby with all the available vaccines or vaccination is done without leaving any child. We have formed greeting/welcome cards for pregnant women. In every ward we go through the list of pregnant women and make them compulsory for four times checkup, to get vaccinated with TT, to take albendazole, iron tablets, to take green leafy vegetables in one of the meals*".

- A 38 years' old health post in-charge from an intervention district

Similarly, investing in human resources like recruiting staff on contract where needed was also observed in both study groups. However, frequency was slightly higher in the intervention districts. "*. . .. previously there was only three staff but now it has increased, because of which we are able to go in the field for growth monitoring*".—A 30 years' old health post in-charge from intervention district.

Contrary to this, in both sorts of districts some stakeholders sought the lack of adequate human resources. Further, in the supply side improvement in the availability of human resources, the regularity of the services, SBA training frequency, medical equipment and other logistics were reported in the intervention districts while these were observed in lower frequency in the comparison districts.

Importantly, in the case of intervention districts unreached population was identified through stakeholder's consultation that led to design and implement targeted interventions. Any such practice was not observed in the comparison districts. Reaching the deprived population was on ad-hoc basis and mainly influenced by non-governmental projects rather than proper planning.

According to the key informants of both intervention and control districts, growing awareness among the community was the main reason for the improvement in mother's and children's health status. The supply side financing as part of the safe motherhood program was mentioned as a motivator for utilization of the health services and institutional delivery among pregnant mothers. Increased in awareness of service utilization (antenatal checkup, institutional delivery, postnatal checkup, immunization and growth monitoring) among stakeholders and beneficiaries was observed in similar ways in both intervention and comparison districts. A VDC secretary credited increased access to communication media for such increased awareness. However, a 50 years' old FCHV from the comparison area said, "*We worked hard in the community to aware golden 1000 days' mothers for service uptake. They are more aware than before. They are also more proactive in seeking information.*"

There were few catchy buy-ins that the local level implemented without a second thought. For instance, incentives for FCHVs upon referring and taking pregnant women to health facilities were adopted by many local levels. In this scheme, FCHVs were given NRs 200–500 per case. This encouragement has helped in ensuring access to otherwise unreached women. Such additional incentives for FCHVs or health workers were rarely found among comparison districts. However, some projects from non-government sectors were providing other types of incentives to them to ensure access of services to all. Similarly, whim of 'zero home delivery' and 'full immunization' had also caught the eye of local leaders and "they agreed easily for such targets", said a health worker from the intervention districts. He added such catchy targets are easily conceived by leaders compared to targets in numbers and percentages.

There were some problems identified by the stakeholders which affected the health services. The problems of chhaupadi, (superstitious beliefs about health) were the key. The issue of being discriminated against by the health workers was also a problem illustrated in the FGDs.

Furthermore, the DIC had been instrumental on making Comprehensive Emergency Obstetrics and Neonatal Care (CEONC) service available and the timely utilization of the services in coordination with FCHVS and health facility staff, according to district level stakeholders. On the other side, even in comparison districts such services had been more accessible than before.

However, some mothers felt no change after the DIC program was implemented while others perceived the changes were visible in terms of increased institutional deliveries, regular mothers' group meetings with the involvement of health workers, availability of 24x7 delivery services, and increased awareness of delivery incentives in the public. Similar changes were also felt by mothers from comparison districts. In backlight, there were some places where access was not ensured due to the lack of the implementation of the plan. On probing, it was found that geographic challenges and limitation of resources kept their foot back.

The people of the Himalayan region stressed on interrupted services due to difficult geography and natural calamity. Stakeholders further added that flooding made access difficult and highlighted the requirement of bridging some of the hurdles of the adequate infrastructure of development such as access to roadways; water supply; flood mitigation; and appropriate health facility buildings. One of the under 2 years' child mother from a comparison district sadly mentioned, "*A child from an affluent family even after spending around five hundred thousand Nepali Rupees and they could not save the child, died of a minor disease like diarrhea, it's so sad*!".

## Discussion

This study examined the perception of local stakeholders regarding health service planning and budgeting and explored the role of the IC approach in influencing the planning and budgeting process to ensure equitable access to MNCH services in Nepal. Four broader themes of descriptions were used, i.e. health planning, resource allocation, implementation of plans and equity in access to MNCH services. Three dimensions of variations were revealed i.e. key aspects, major focus and impact of the DIC.

Improvement in MNCH health service coverage was found to be similar in both groups. However, subjective information suggests that the DIC approach has helped in improving coverage and equity in access in the intervention districts. Data from the Health Management Information System over the years suggest similar trends of improvement in coverage [18, 24] in intervention and non-intervention districts. One reason of the similarity in both groups could be the contamination of knowledge across districts through media or staff transfer. Also, central government driven programs and approaches in combination with efforts from various non-government actors through different projects have well influenced health coverage in both groups. This could also be partially a result of low-quality data at lower level.

The study showed that the government stakeholders found the investment case approach useful to identify problems/challenges using evidence-based data and found solutions to address the issues related to MNCH. The IC approach has helped to improve positive acceptance and the approach is adopted by the community level stakeholders. A study conducted for Evaluation of Accelerating the Implementation of the Investment Case for Maternal, Newborn and Child Health in Asia and the Pacific Programme concluded that the IC approach brought about positive changes and the indicators are showing upward trends [25]. Females especially FCHVs, were also found to be involved in planning process in some of the IC implemented areas. However, perspectives from community stakeholders showed that there is a lack of equal involvement of representatives from different sectors. During the planning process only few of them had the opportunity to be involved, whereas others were only informed

about meetings and their outcomes. Evidence shows that engagement of the community brings about equitable and high-quality MNCH services [26]. Empowering women on health and economic aspects has shown to have positive impact on improving awareness and utilizing maternal, neonatal and child health related services as well as uplifting livelihood of the community people [27].

Most of the stakeholders in the intervention districts knew about the IC approach. According to the stakeholders, the IC approach has played an effective role in systemic planning and execution of health activities. According to the IC approach, all stakeholders participate in the planning process. This is supported by a report of UNICEF on IC impact evaluation report of Nepal and Indonesia in which planning, and budgeting started after implementing the IC approach [28].

Considering the budgetary aspects, the budget for MNCH related activities has increased according to the stakeholders. In comparison to the previous health plan, the budget has risen and the VDC also has started to recognize its importance. This could be reached mainly by the involvement of related stakeholders in health-planning workshops and by this, making them accountable for the health service development. Investment in fundamental resources such as infrastructure and staff are critical which was revealed by an IC analysis in India [14]. An estimation of the budget is a useful aspect of the IC approach which helps to guide towards the best decisions among various strategies [13]. The IC approach solely focuses on maternal, neonatal, child health and nutrition. From an IC evaluation study, the district health managers recommended for comprehensive planning approach which would cover other health areas too [25]. The comparison districts lacked this kind of proper planning and budgeting.

Inter-sectoral coordination is crucial to achieve any health-related indicators [29]. The stakeholders also mentioned the inter-sectoral collaboration between health, nutrition, sanitation, agriculture, women and children welfare sectors. In addition, the health workers also coordinated with development partners like UNICEF, USAID and other non-governmental partners.

Coming to the supply side, proper financing shows improvement of the quality of health facilities like the availability of skilled health workers, the regularity in services and the expansion of the range of the services. The same idea is provided by the Tanahashi model which states that the availability of manpower, facilities and drugs maximizes the capacity of the services [15]. According to the stakeholders from the intervention districts, the major health improvement was in the sector of Maternal and Child Health like antenatal care, institutional delivery and child immunization which was due to the availability of health workers in the health facilities. A project named "The East Africa Maternal Newborn Child Health project (EAMNeCH)" was implemented in four vulnerable countries of Africa which focused on quality, demand, supply of maternal, neonatal and child health services and favorable policy environment. The results of the project contributed toward also improvement in MNCH indicators [30]. Health workers mentioned the filling of vacant posts, regularity and access of the services in the intervention districts which became key to the improvement of MNCH indicators. According to a systematic review conducted in low-income countries in Africa also supported the findings in which lack of skilled health services providers and access to health facilities were the barriers to utilization of MNCH services [31]. Other mentioned reasons towards the improvement of MNCH related indicators were an increased access to communication and a positive attitude of pregnant mothers and their in-laws towards the health service. Studies have shown that with the adequate availability of skilled providers, an antenatal care coverage of over 60% can be reached [32]. The same result has been seen in the Philippines, where a study concluded that access is one key factor to improve service utilization.

Research in Indonesia also found that cities with better access were receiving quality care [14]. Other studies also have identified access to increase service utilization. The higher utilization was seen in those who lived near to health facilities in comparison to those living far away [33, 34]. This was also supported by a case study conducted in Nepal. There was increase in institutional delivery by establishing birthing centers near to the community [35]. Achieving equity in health among marginalized people will only increase by improving access to them specifically, facing the most disadvantaged [36]. Service-seeking behavior has a direct influence on the acceptance of health programs of a population; thus, alternative care influences the service utilization patterns like family planning, ANC, institutional deliveries, PNC, and vaccine services. An increase in awareness among community people has a direct impact on positive outcomes in behaviors like exclusive breastfeeding and umbilical cord care [32, 35].

Stakeholders stated that the geographical location is critical for the uptake of the utilization of health services and varies in the different seasons; mainly in the Himalayan and the Terai districts. Difficult physical access to health services due to geographical difficulty has been a major challenge for maternal health service utilization [37, 38]. Likewise, it has been observed that a permanent method for family planning has been most likely used in the Terai region (23%) compared to the hill districts (12%). Besides, there were other challenges including frequent transfer of staff, the inadequacy of staff housing, insufficient medicines and supplies which affected the utilization and quality of services. The presence of cultural taboos and superstitious beliefs about health influenced the utilization of services. The other prominent issues were language barrier and social inclusiveness, which affected the quality of health care adversely [32].

The IC approaches main positive aspect felt by stakeholders was its inclusiveness, the process included a variety of participants like health managers, health workers, related stakeholders from different sectors, politicians including the local leaders (in political arena), civil society and media people which made it unique. The stakeholders of Bangladesh also revealed that the IC approach has empowered the stakeholders to make planning and decisions in the health sector and has provided insights to improve MNCH [25].

The rate of current improvement in health service use is not enough to achieve the Sustainable Development Goals (SDGs) because the targets ahead are difficult to reach. Those people who are out of coverage are often from hard to reach population. For instance, Nepal has achieved 55.6% reduction in maternal mortality ratio (539 to 239 maternal deaths per 100,000 live births) between 1996 to 2015 [9, 39]. However, the current rate of reduction is far behind a required nearly 91% decrease to achieve the SDG target of only 20 maternal deaths per 100,000 live births by 2030 [40].

Few things need to be considered to maximize the effect of the IC process. The IC approach should include a systematic monitoring and feedback system. It was realized that the expansion of birthing centers, ensuring regular supplies, fulfillment of sanctioned (government allocated) post would further enhance MNCH outcomes [14]. The monitoring and feedback system had been introduced and had been also stressed by the stakeholders in the IC districts where a decline in maternal, neonate and child mortality could be observed. It was felt that strong advocacy for the commitment from the government and community people should be assured. The ownership of the plans by community people and stakeholders were felt imperative for the successful implementation of these plans [14, 35]. Case studies from Mozambique, Nepal and Rwanda have shown that improved health financing, decentralization of decision making and service delivery, task-shifting, development of partnerships, adequate Community Health Workers and community engagement were the important drivers for uplifting maternal and child health [41].

The study has some limitations: the study was only conducted in 6 districts (3 intervention districts and 3 control districts) out of 21 intervention and 56 comparison districts. Purposive sampling method was used in the study to select the districts so findings may not be generalized to all IC implemented and non-implemented districts. There may be possibility of selection bias as the mothers of under 2 years' children and health workers were selected as per the convenience. Data were collected in Nepali language and were translated into English for analysis, which may have caused loss in some loss of meanings of some important issues.

## Conclusion

The findings of this research provide insights into and information about the practices associated with health planning and the influence of the DIC approach in ensuring equity in maternal and child health services. There was a unanimous voice among stakeholders that health status has improved including nutritional status in the districts where the IC approach was implemented. The stakeholders unanimously felt the improvement of access to services with the increase in the number of birthing centers in the IC approach districts and an improvement of the budget for health-related activities encouraging. The increased access to communication, positive attitude of pregnant mothers towards health service utilization and the support of family and access are contributing factors to improve the health of mother, neonate, and child. The district stakeholders felt a decline in maternal mortality, neonate, and child mortality that shows positive deviance of health sector improvement in maternal, neonatal and child health in IC districts compared to control districts. It was seen that districts without IC showed numerous flaws on implementation. Major challenges were lack of ownership, top down planning, dilapidated infrastructure, irregular supply, demotivating staff, resulting in despaired outcome. It is also recommended to implement similar or such type of approach in other non-intervention districts with strong supervision and monitoring in order to strengthen planning and budgeting in MNCH related programs.

Though the IC approach has numerous positive outcomes, it is recommended that the process focuses on the inclusiveness of the planning and to strengthening the monitoring component by several stakeholders.

## Supporting information

**S1 Raw data.**
(DOCX)

## Acknowledgments

We would like to acknowledge the support received from UNICEF Nepal, Little Buddha College of Health Science, Mr. Dip Narayan Thakur, Mr. Raj Kumar Subedi, Ms. Monica Manandhar, and Ms. Radhika Sapkota for their support during the study.

## Author Contributions

**Conceptualization:** Janak Kumar Thapa, Doris Stöckl, Mahesh Kumar Maskey, Maria Delius.

**Data curation:** Janak Kumar Thapa, Raj Kumar Sangroula, Meena Thapa.

**Formal analysis:** Janak Kumar Thapa, Raj Kumar Sangroula.

**Investigation:** Janak Kumar Thapa.

**Methodology:** Janak Kumar Thapa, Raj Kumar Sangroula, Maria Delius.

**Project administration:** Janak Kumar Thapa.

**Resources:** Raj Kumar Sangroula, Asha Pun, Meena Thapa.

**Software:** Janak Kumar Thapa, Raj Kumar Sangroula.

**Supervision:** Doris Stöckl, Mahesh Kumar Maskey, Maria Delius.

**Validation:** Asha Pun.

**Visualization:** Janak Kumar Thapa, Raj Kumar Sangroula, Maria Delius.

**Writing – original draft:** Janak Kumar Thapa.

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
