## [Decision Letter · Decision Letter 0]

16 Feb 2021

PONE-D-20-40391

Investment Case Approach for Equitable Access to Maternal Neonatal and Child Health Services: Stakeholders’ Perspective in Nepal.

PLOS ONE

Dear Dr. Thapa,

Thank you for submitting your manuscript to PLOS ONE. After careful consideration, we feel that it has merit but does not fully meet PLOS ONE’s publication criteria as it currently stands. Therefore, we invite you to submit a revised version of the manuscript that addresses the points raised during the review process.

We look forward to receiving your revised manuscript.

Kind regards,

Bidhubhusan Mahapatra, Ph.D.

Academic Editor

PLOS ONE

Journal Requirements:

2.We note that you have indicated that data from this study are available upon request. PLOS only allows data to be available upon request if there are legal or ethical restrictions on sharing data publicly. For information on unacceptable data access restrictions, please see http://journals.plos.org/plosone/s/data-availability#loc-unacceptable-data-access-restrictions.

3.Thank you for stating the following financial disclosure:

"NO - Include this sentence at the end of your statement: The funders had no role in study design, data collection and analysis, decision to publish, or preparation of the manuscript."

Additional Editor Comments:

It is an interesting piece of work. My concern is related to the relevance of the study given the data used was collected way back in 2014. There needs to be justification around using such an old data, specifically, why it is still relevant in the context of Nepal given the country has gone through major changes since 2014. Even the introduction needs revision in terms of citing recent literature and building the right context. As both the reviewer suggests, the paper needs thorough English language editing.

Reviewers' comments:

Reviewer's Responses to Questions

**Comments to the Author**

1. Is the manuscript technically sound, and do the data support the conclusions?

Reviewer #1: Yes

Reviewer #2: Yes

2. Has the statistical analysis been performed appropriately and rigorously? 

Reviewer #1: N/A

Reviewer #2: N/A

3. Have the authors made all data underlying the findings in their manuscript fully available?

Reviewer #1: Yes

Reviewer #2: Yes

4. Is the manuscript presented in an intelligible fashion and written in standard English?

Reviewer #1: No

Reviewer #2: Yes

5. Review Comments to the Author

Reviewer #1: It is an interesting piece to read and authors have tried to cover all the aspects of the DIC program. However, it feels the introduction section can be more tightly written, to give the reader and idea of what to expect in the paper in first 2 paragraphs. Thorough English editing is needed, and too much citation of past work in the data section instead of the original work done by the authors. Few detailed observations are given below-

The first line in introduction is about MDGs, isn’t it’s too late to talk about MDG goals? Why not state and compare where Nepal is as per SDG goals which is reducing it to less than 70 per hundred thousand live births.

On page 3 the sentence says “In this regard, it is important to have actions to achieve higher quality at local level……” author needs to elaborate a little higher quality in what?

Provide reference for the sentence “The knowledge of evidence-based approaches and effective frameworks for maternal health management is low among local stakeholders.”

Better to provide examples on who all can be stakeholders in the introduction itself.

In study participants- describe who is program implementers. And role of each key stakeholders in the ongoing DIC program.

Data collections section needs to be rewritten with more clarity and details like saying only interviews is not enough, authors should mention whether its face to face interviews, who is conducting interview? Whether authors themselves did it? Who did the translation, validity of translation? Who all was involved in the analysis?

On page 11 what does KII means? Need to give the full form.

What was the division of sample between intervention and control districts? Was it same or different?

Data section part is very poorly written, most of the sentences are taken from some other studies instead of writing what was done for the present study. It will be better to note down all the steps for clarity of the readers in case someone wants to replicate this approach in future. For example- on page 12 instead of writing that “researchers’ interpretations were controlled and cross-checked….” Please elaborate what was done to control and cross-check?

On page 12 it is mentioned “adopted an analytical framework of structure of perception and experiences……….” Which framework is this? No details are given about this in the paper.

In second paragraph of the discussion authors have written “knowledge across nations through media or staff transfer….” Shouldn’t it be knowledge across districts?

On page 22, “Females, marginalized people and disabled people were also found to be involved, however this is not confirmed by literature”. Why this needs to be confirmed by literature? If this is a new program, and the present study is an evaluation authors should state their findings with confidence without worrying about literature. Moreover, this sentence feels contradictory to what is mentioned in the results- “women in both study groups, that they or their family members are often not consulted during the village level annual MNCH planning”.

Looks like there is discrepancy in what women are saying and what other stakeholders are saying regarding involvement of village population. A focus on this should be added in the discussion.

Reviewer #2: Thanks for giving me the opportunity to review the paper. Please see my comments as below.

Study setting: The study was conducted in in June 2014 in purposively selected

districts based on HDI values, three intervention and three comparison districts. Delete one “in” in the first sentence.

In the method section: Please add a brief paragraph on data collection specifying: Who were the FGD facilitators (what were their qualifications, what training did they receive in FGD facilitation)? Was there a note-taker? Where did the interviews take place? What was the consent process? Were the information collected in of KIIs and FGDs different?

In the analysis section, please specify the following: How was the coding done on full transcripts? How many people were involved in this? How was the iteration done? Was any double-coding done to check inter-coder reliability? How have you checked the interpretation validity and descriptive validity?

Did you combine findings from KIIs and FGDs while presenting the results?

In the results section, please present some background info of the participants. After each quote please write some info of the participants: age, sex, years of experience, designation, intervention or comparison group.

Discussion:

In general, discussion needs to be better situated in the literature. Look for other studies of interventions or program/policies that might have worked to address your specific issues, even from other countries.

What are the limitations of the study? Please describe in the text.

Conclusion: “It was seen that districts without IC showed numerous flaws on implementation”. What would be your recommendation for them?

Finally, the manuscript needs editing by a native English speaker. There are many inconsistencies in grammar throughout the manuscript.

6. PLOS authors have the option to publish the peer review history of their article (what does this mean?). If published, this will include your full peer review and any attached files.

Reviewer #1: **Yes: **Anktia Shukla

Reviewer #2: No

---

## [Author Response · Author response to Decision Letter 0]

5 May 2021

Response to Reviewer 1

Reviewer #1: It is an interesting piece to read and authors have tried to cover all the aspects of the DIC program. However, it feels the introduction section can be more tightly written, to give the reader and idea of what to expect in the paper in the first 2 paragraphs. Thorough English editing is needed, and too much citation of past work in the data section instead of the original work done by the authors. Few detailed observations are given below-

Response: Thank You for your valuable comments which mean a lot for us. We have edited the introduction section and rewritten the portion.

Comment: The first line in introduction is about MDGs, isn’t it’s too late to talk about MDG goals? Why not state and compare where Nepal is as per SDG goals which is reducing it to less than 70 per hundred thousand live births.

Response: Thank you for your comment. We have edited the sentences as, “Nepal has achieved several Millennium Development Goals (MDGs) and has been close to attaining the target of reducing maternal mortality by 75% from 258 to 213 per 100,000 live births (1) and is progressing towards achieving the SDG target of 70/100,000 live birth (2)” – page number 2

Comment: On page 3 the sentence says “In this regard, it is important to have actions to achieve higher quality at local level……” author needs to elaborate a little higher quality in what?

Provide reference for the sentence “The knowledge of evidence-based approaches and effective frameworks for maternal health management is low among local stakeholders.”

Better to provide examples on who all can be stakeholders in the introduction itself..

Response: Thank you for your comments. We have edited the introduction portion. (page number 2,3 and 4). We have also mentioned the local stakeholders in the introduction portion as The main actors for implementation of the DIC programs were District Health Officers, Public Health Nurses, Local Development Officers, VDC secretary, frontline Health workers from the community level, FCHVs and so on –page number 3

Comment: In study participants- describe who is program implementers. And role of each key stakeholders in the ongoing DIC program.

Response: Thank you for your comments. We have re-written the study participants in the methods section. – page number 5 and 6

Comments: Data collections section needs to be rewritten with more clarity and details like saying only interviews is not enough, authors should mention whether its face to face interviews, who is conducting interview? Whether authors themselves did it? Who did the translation, validity of translation? Who all was involved in the analysis?

Response: Thank you for your valuable comments. We have re-written the data collection section in method part of the manuscript. – page number 7 and 8

Comment: On page 11 what does KII means? Need to give the full form. 

Response: Thank you. we have mentioned the full form of KII as Key Informant Interview in the manuscript.

Comment: What was the division of sample between intervention and control districts? Was it same or different?

Response: The sample size both the intervention and comparison districts were same and level of the study participants were also same. We have included it in the methods section.- page number 5

Comment: Data section part is very poorly written, most of the sentences are taken from some other studies instead of writing what was done for the present study. It will be better to note down all the steps for clarity of the readers in case someone wants to replicate this approach in future. For example- on page 12 instead of writing that “researchers’ interpretations were controlled and cross-checked….” Please elaborate what was done to control and cross-check?

Response: Thank you. We have re-written the part. – page number 6,7 and 8

Comment: On page 12 it is mentioned “adopted an analytical framework of structure of perception and experiences……….” Which framework is this? No details are given about this in the paper.

Response: Thank you. we have omitted the sentence. The part is re-written. – page number 9

Comment: In second paragraph of the discussion authors have written “knowledge across nations through media or staff transfer….” Shouldn’t it be knowledge across districts?

Response: Thank you. We have edited the sentence. – page number 19

Comments: On page 22, “Females, marginalized people and disabled people were also found to be involved, however this is not confirmed by literature”. Why this needs to be confirmed by literature? If this is a new program, and the present study is an evaluation authors should state their findings with confidence without worrying about literature. Moreover, this sentence feels contradictory to what is mentioned in the results- “women in both study groups, that they or their family members are often not consulted during the village level annual MNCH planning”.

Looks like there is discrepancy in what women are saying and what other stakeholders are saying regarding involvement of village population. A focus on this should be added in the discussion.

Response: Thank you. We have edited the discussion part as commented and we have added some literatures in the discussion part as well.

 

Responses to reviewer-2 

Comment: Study setting: The study was conducted in in June 2014 in purposively selected

districts based on HDI values, three intervention and three comparison districts. Delete one “in” in the first sentence.

Response: Thank you for your valuable comment. we have rearranged the sentence

Comment: In the method section: Please add a brief paragraph on data collection specifying: Who were the FGD facilitators (what were their qualifications, what training did they receive in FGD facilitation)? Was there a note-taker? Where did the interviews take place? What was the consent process? Were the information collected in of KIIs and FGDs different? 

Response: Thank you for your valuable comment. For the collection of data, 3 FGD facilitators with Bachelor’s in Public Health degree with 2 years’ experience and three note takers with Bachelor’s degree in Public Health with 2 years’ experience in qualitative research were recruited. The same teams were involved in taking Key-informant interviews too. Three days’ intensive qualitative research-related training was given to the team by the lead researchers and other co-researchers who were expert in qualitative research. The training consisted of a different session on qualitative research including an introduction to qualitative research, scope, approaches, methods, orientation to the tools, discussion, mock session and exercise in real field setting. The key-informant interviews took place in the District Health Offices, Local Development Offices, Health Posts, Districts Hospitals of the respective districts. The themes of the information collected from KIIs and FGDs were same. Both included the impact of Investment Case approach on maternal and child health status.

– page number 8

Comment: In the analysis section, please specify the following: How was the coding done on full transcripts? How many people were involved in this? How was the iteration done? Was any double-coding done to check inter-coder reliability? How have you checked the interpretation validity and descriptive validity? 

Response: Thank you for your valuable comment. In the “analysis of the data” and Rigor, Reliability, and Validity part of the method section, we have edited the part as: We highlighted various phrases in different colors corresponding to different codes in which each code described the idea or feeling expressed in that part of the text. – page number 8 

For coding and making themes from the codes, the lead researcher was mainly involved. The co-researchers assisted in the process. For all the steps of data analysis, the lead researcher was mostly involved and in discussion with the co-researchers, the analysis was finalized. All the co-authors agreed- following the discussion while finalizing the codes and the themes. – page number 9

Study tools were finalized in consultation with the qualitative as well as subject experts, and the data were collected by well-trained field researchers. – page number 9

The collected data were checked by the lead researcher and other co-researchers for their accuracy and quality for ensuring descriptive and interpretation validity. In case of missing data, the participants were followed up on the next day. – page number 9

Did you combine findings from KIIs and FGDs while presenting the results? 

Response: Thank you for your valuable comment. we have combined the finding from KIIs and FGDs while presenting the results.

In the results section, please present some background info of the participants. After each quote please write some info of the participants: age, sex, years of experience, designation, intervention or comparison group. 

Response: Thank you for your valuable comment. We have revised the sections as per the comments given.

Discussion: 

Comments: In general, discussion needs to be better situated in the literature. Look for other studies of interventions or program/policies that might have worked to address your specific issues, even from other countries. 

Response: Thank you for your valuable comments. We have added related literatures for discussion. They appear as: 

A study conducted for Evaluation of Accelerating the Implementation of the Investment Case for Maternal, Newborn and Child Health in Asia and the Pacific Programme concluded that IC approach brought about positive changes and the indicators are showing upward trends - page number 19

The IC approach solely focuses on maternal, neonatal, child health and nutrition. From an IC evaluation study, the district health managers recommended for comprehensive planning approach which would cover other health areas too- page number 20,21

A project named “The East Africa Maternal Newborn Child Health project (EAMNeCH)” was implemented in four vulnerable countries of Africa which focused on quality, demand, supply of maternal, neonatal and child health services and favorable policy environment. The results of the project contributed toward also improvement in MNCH indicators - page number 21

According to a systematic review conducted in low income countries in Africa also supported the findings in which lack of skilled health services providers and access to health facilities were the barriers to utilization of MNCH services - page number 21

This was also supported by a case study conducted in Nepal. There was increase in institutional delivery by establishing birthing centers near to the community - page number 22

Case studies from Mozambique, Nepal and Rwanda have shown that improved health financing, decentralization of decision making and service delivery, task-shifting, development of partnerships, adequate Community Health Workers and community engagement were the important drivers for uplifting maternal and child health - page number 24

Comment: What are the limitations of the study? Please describe in the text. 

The study has some limitation: the study was only conducted in 6 districts (3 intervention districts and 3 control districts) out of 21 intervention and 56 comparison districts. - Page number 21

Conclusion: “It was seen that districts without IC showed numerous flaws on implementation”. What would be your recommendation for them?

Response: We have added the part as “It is also recommended to implement similar or such type of approach in other non-intervention districts with strong supervision and monitoring in order to strengthen planning and budgeting in MNCH related programs. – page number 25

Comment: Finally, the manuscript needs editing by a native English speaker. There are many inconsistencies in grammar throughout the manuscript. 

Response: Thank you for your valuable comment. The manuscript has been edited by a native English speaker.

---

## [Editor Report · Decision Letter 1]

9 May 2021

PONE-D-20-40391R1

Investment Case Approach for Equitable Access to Maternal, Neonatal and Child Health Services: Stakeholders’ Perspective in Nepal.

PLOS ONE

Dear Dr. Thapa,

Thank you for submitting your manuscript to PLOS ONE. After careful consideration, we feel that it has merit but does not fully meet PLOS ONE’s publication criteria as it currently stands. Therefore, we invite you to submit a revised version of the manuscript that addresses the points raised during the review process.

We look forward to receiving your revised manuscript.

Kind regards,

Bidhubhusan Mahapatra, Ph.D.

Academic Editor

PLOS ONE

Additional Editor Comments (if provided):

Dear Author, I see that the you have not responded to the editor's comments. Also, the track version does not reflect the changes you made. Therefore, returning this paper to make necessary changes in track mode and responding to editor's comments.

---

## [Author Response · Author response to Decision Letter 1]

15 May 2021

I have to mention in Manuscript and revised Manuscript with Track Changes'. Also mention in Response to Reviewers.

---

## [Editor Report · Decision Letter 2]

20 May 2021

PONE-D-20-40391R2

Investment Case Approach for Equitable Access to Maternal, Neonatal and Child Health Services: Stakeholders’ Perspective in Nepal.

PLOS ONE

Dear Dr. Thapa,

Thank you for submitting your manuscript to PLOS ONE. After careful consideration, we feel that it has merit but does not fully meet PLOS ONE’s publication criteria as it currently stands. Therefore, we invite you to submit a revised version of the manuscript that addresses the points raised during the review process.

We look forward to receiving your revised manuscript.

Kind regards,

Bidhubhusan Mahapatra, Ph.D.

Academic Editor

PLOS ONE

Additional Editor Comments (if provided):

Thank you for uploading the track version of the manuscript. While several of the comments are addressed, I feel some of the important aspects are yet to be addressed completely. Reviewer 2 had suggested to provide more details on the method of data collection, analysis and highlight the study limitations. While the authors have provided some more details related to the data collection and analysis, I feel it is still not sufficient. I suggest authors to follow the COREQ checklist, or other relevant checklists listed by the Equator Network, such as the SRQR, to ensure complete reporting (http://journals.plos.org/plosone/s/submission-guidelines#loc-qualitative-research). In general, it is expected that qualitative studies to include the following: 1) defined objectives or research questions; 2) description of the sampling strategy, including rationale for the recruitment method, participant inclusion/exclusion criteria and the number of participants recruited; 3) detailed reporting of the data collection procedures; 4) data analysis procedures described in sufficient detail to enable replication; 5) a discussion of potential sources of bias; and 6) a discussion of limitations. Currently, your manuscript does not have sufficient details and lacks clarity how the coding was performed.

In my first round of review, I had asked authors to include justification around the use of data from 2014 which is quite old now. However, it has not been addressed yet.

Finally, the manuscript has several limitations. Authors should clearly list them out. It is important that a good paper also highlights the limitations associated with the data and helps readers in making right inference from the paper.

---

## [Author Response · Author response to Decision Letter 2]

8 Jul 2021

I have addressed all reviewer comments and attached in the portal.

---

## [Editor Report · Decision Letter 3]

13 Jul 2021

Investment Case Approach for Equitable Access to Maternal, Neonatal and Child Health Services: Stakeholders’ Perspective in Nepal. 

PONE-D-20-40391R3

Dear Dr. Thapa,

We’re pleased to inform you that your manuscript has been judged scientifically suitable for publication and will be formally accepted for publication once it meets all outstanding technical requirements.

Kind regards,

Bidhubhusan Mahapatra, Ph.D.

Academic Editor

PLOS ONE
---

## [Editor Report · Acceptance letter]

20 Sep 2021

PONE-D-20-40391R3 

Investment Case Approach for Equitable Access to Maternal Neonatal and Child Health Services: Stakeholders’ Perspective in Nepal 

Dear Dr. Thapa:

I'm pleased to inform you that your manuscript has been deemed suitable for publication in PLOS ONE. Congratulations! Your manuscript is now with our production department. 

Kind regards, 

on behalf of

Dr. Bidhubhusan Mahapatra 

Academic Editor

PLOS ONE